# Trellis Networks for Sequence Modeling

**Shaojie Bai**
Carnegie Mellon University

**J. Zico Kolter**
Carnegie Mellon University and
Bosch Center for AI

**Vladlen Koltun**
Intel Labs

## Abstract

We present trellis networks, a new architecture for sequence modeling. On the one hand, a trellis network is a temporal convolutional network with special structure, characterized by weight tying across depth and direct injection of the input into deep layers. On the other hand, we show that truncated recurrent networks are equivalent to trellis networks with special sparsity structure in their weight matrices. Thus trellis networks with general weight matrices generalize truncated recurrent networks. We leverage these connections to design high-performing trellis networks that absorb structural and algorithmic elements from both recurrent and convolutional models. Experiments demonstrate that trellis networks outperform the current state of the art methods on a variety of challenging benchmarks, including word-level language modeling and character-level language modeling tasks, and stress tests designed to evaluate long-term memory retention. The code is available here[1].

## 1 Introduction

What is the best architecture for sequence modeling? Recent research has produced significant progress on multiple fronts. Recurrent networks, such as LSTMs, continue to be optimized and extended (Merity et al., 2018b; Melis et al., 2018; Yang et al., 2018; Trinh et al., 2018). Temporal convolutional networks have demonstrated impressive performance, particularly in modeling long-range context (van den Oord et al., 2016; Dauphin et al., 2017; Bai et al., 2018). And architectures based on self-attention are gaining ground (Vaswani et al., 2017; Santoro et al., 2018).

In this paper, we introduce a new architecture for sequence modeling, the Trellis Network. We aim to both improve empirical performance on sequence modeling benchmarks and shed light on the relationship between two existing model families: recurrent and convolutional networks.

On the one hand, a trellis network is a special temporal convolutional network, distinguished by two unusual characteristics. First, the weights are tied across layers. That is, weights are shared not only by all time steps but also by all network layers, tying them into a regular trellis pattern. Second, the input is injected into all network layers. That is, the input at a given time-step is provided not only to the first layer, but directly to all layers in the network. So far, this may seem merely as a peculiar convolutional network for processing sequences, and not one that would be expected to perform particularly well.

Yet on the other hand, we show that trellis networks generalize truncated recurrent networks (recurrent networks with bounded memory horizon). The precise derivation of this connection is one of the key contributions of our work. It allows trellis networks to serve as bridge between recurrent and convolutional architectures, benefitting from algorithmic and architectural techniques developed in either context. We leverage these relationships to design high-performing trellis networks that absorb ideas from both architectural families. Beyond immediate empirical gains, these connections may serve as a step towards unification in sequence modeling.

We evaluate trellis networks on challenging benchmarks, including word-level language modeling on the standard Penn Treebank (PTB) and the much larger WikiText-103 (WT103) datasets; character-level language modeling on Penn Treebank; and standard stress tests (e.g. sequential MNIST, permuted MNIST, etc.) designed to evaluate long-term memory retention. On word-level

---

[1] https://github.com/locuslab/trellisnet

Penn Treebank, a trellis network outperforms by more than a unit of perplexity the recent architecture search work of Pham et al. (2018), as well as the recent results of Melis et al. (2018), which leveraged the Google Vizier service for exhaustive hyperparameter search. On character-level Penn Treebank, a trellis network outperforms the thorough optimization work of Merity et al. (2018a). On word-level WikiText-103, a trellis network outperforms by 7.6% in perplexity the contemporaneous self-attention-based Relational Memory Core (Santoro et al., 2018), and by 11.5% the work of Merity et al. (2018a). (Concurrently with our work, Dai et al. (2019) employ a transformer and achieve even better results on WikiText-103.) On stress tests, trellis networks outperform recent results achieved by recurrent networks and self-attention (Trinh et al., 2018). It is notable that the prior state of the art across these benchmarks was held by models with sometimes dramatic mutual differences.

## 2 BACKGROUND

Recurrent networks (Elman, 1990; Werbos, 1990; Graves, 2012), particularly with gated cells such as LSTMs (Hochreiter & Schmidhuber, 1997) and GRUs (Cho et al., 2014), are perhaps the most popular architecture for modeling temporal sequences. Recurrent architectures have been used to achieve breakthrough results in natural language processing and other domains (Sutskever et al., 2011; Graves, 2013; Sutskever et al., 2014; Bahdanau et al., 2015; Vinyals et al., 2015; Karpathy & Li, 2015). Convolutional networks have also been widely used for sequence processing (Waibel et al., 1989; Collobert et al., 2011). Recent work indicates that convolutional networks are effective on a variety of sequence modeling tasks, particularly ones that demand long-range information propagation (van den Oord et al., 2016; Kalchbrenner et al., 2016; Dauphin et al., 2017; Gehring et al., 2017; Bai et al., 2018). A third notable approach to sequence processing that has recently gained ground is based on self-attention (Vaswani et al., 2017; Santoro et al., 2018; Chen et al., 2018). Our work is most closely related to the first two approaches. In particular, we establish a strong connection between recurrent and convolutional networks and introduce a model that serves as a bridge between the two. A related recent theoretical investigation showed that under a certain stability condition, recurrent networks can be well-approximated by feed-forward models (Miller & Hardt, 2018).

There have been many combinations of convolutional and recurrent networks (Sainath et al., 2015). For example, convolutional LSTMs combine convolutional and recurrent units (Donahue et al., 2015; Venugopalan et al., 2015; Shi et al., 2015). Quasi-recurrent neural networks interleave convolutional and recurrent layers (Bradbury et al., 2017). Techniques introduced for convolutional networks, such as dilation, have been applied to RNNs (Chang et al., 2017). Our work establishes a deeper connection, deriving a direct mapping across the two architectural families and providing a structural bridge that can incorporate techniques from both sides.

## 3 SEQUENCE MODELING AND TRELLIS NETWORKS

**Sequence modeling.** Given an input $x_{1:T} = x_1, \ldots, x_T$ with sequence length $T$, a sequence model is any function $G : \mathcal{X}^T \to \mathcal{Y}^T$ such that

$$y_{1:T} = y_1, \ldots, y_T = G(x_1, \ldots, x_T), \tag{1}$$

where $y_t$ should only depend on $x_{1:t}$ and not on $x_{t+1:T}$ (i.e. no leakage of information from the future). This causality constraint is essential for autoregressive modeling.

In this section, we describe a new architecture for sequence modeling, referred to as a trellis network or TrellisNet. In particular, we provide an atomic view of TrellisNet, present its fundamental features, and highlight the relationship to convolutional networks. Section 4 will then elaborate on the relationship of trellis networks to convolutional and recurrent models.

**Notation.** We use $x_{1:T} = (x_1, \ldots, x_T)$ to denote a length-$T$ input sequence, where vector $x_t \in \mathbb{R}^p$ is the input at time step $t$. Thus $x_{1:T} \in \mathbb{R}^{T \times p}$. We use $z_t^{(i)} \in \mathbb{R}^q$ to represent the hidden unit at time $t$ in layer $i$ of the network. We use $\text{Conv1D}(x; W)$ to denote a 1D convolution with a kernel $W$ applied to input $x = x_{1:T}$.

**A basic trellis network.** At the most basic level, a feature vector $z_{t+1}^{(i+1)}$ at time step $t + 1$ and level $i + 1$ of TrellisNet is computed via three steps, illustrated in Figure 1a:

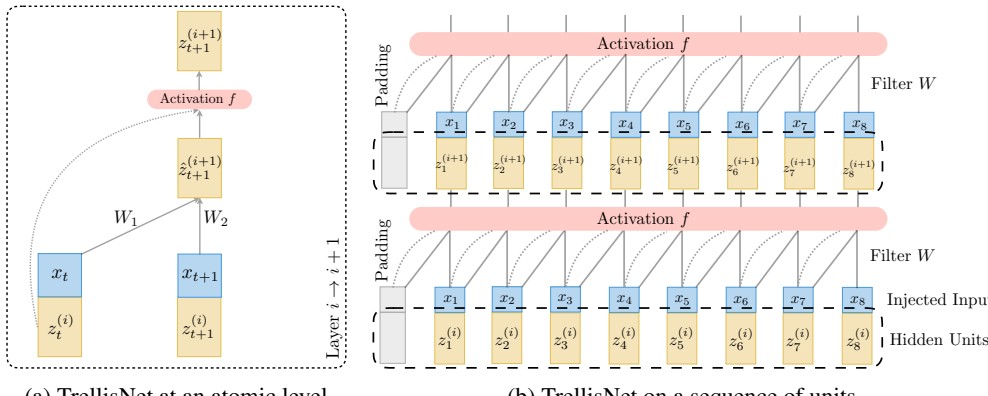

| (a) TrellisNet at an atomic level | (b) TrellisNet on a sequence of units |

Figure 1: The interlayer transformation of TrellisNet, at an atomic level (time steps $t$ and $t+1$, layers $i$ and $i+1$) and on a longer sequence (time steps 1 to 8, layers $i$ and $i+1$).

1. The hidden input comprises the hidden outputs $z_t^{(i)}, z_{t+1}^{(i)} \in \mathbb{R}^q$ from the previous layer $i$, as well as an injection of the input vectors $x_t, x_{t+1}$. At level 0, we initialize to $z_t^{(0)} = \mathbf{0}$.

2. A pre-activation output $\hat{z}_{t+1}^{(i+1)} \in \mathbb{R}^r$ is produced by a feed-forward linear transformation:

$$\hat{z}_{t+1}^{(i+1)} = W_1 \begin{bmatrix} x_t \\ z_t^{(i)} \end{bmatrix} + W_2 \begin{bmatrix} x_{t+1} \\ z_{t+1}^{(i)} \end{bmatrix}, \tag{2}$$

where $W_1, W_2 \in \mathbb{R}^{r \times (p+q)}$ are weights, and $r$ is the size of the pre-activation output $\hat{z}_{t+1}^{(i+1)}$. (Here and throughout the paper, all linear transformations can include additive biases. We omit these for clarity.)

3. The output $z_{t+1}^{(i+1)}$ is produced by a nonlinear activation function $f : \mathbb{R}^r \times \mathbb{R}^q \to \mathbb{R}^q$ applied to the pre-activation output $\hat{z}_{t+1}^{(i+1)}$ and the output $z_t^{(i)}$ from the previous layer. More formally, $z_{t+1}^{(i+1)} = f\left(\hat{z}_{t+1}^{(i+1)}, z_t^{(i)}\right)$.

A full trellis network can be built by tiling this elementary procedure across time and depth. Given an input sequence $x_{1:T}$, we apply the same production procedure across all time steps and all layers, using the same weights. The transformation is the same for all elements in the temporal dimension and in the depth dimension. This is illustrated in Figure 1b. Note that since we inject the same input sequence at every layer of the TrellisNet, we can precompute the linear transformation $\tilde{x}_{t+1} = W_1^x x_t + W_2^x x_{t+1}$ for all layers $i$. This identical linear combination of the input can then be added in each layer $i$ to the appropriate linear combination of the hidden units, $W_1^z z_t^{(i)} + W_2^z z_{t+1}^{(i)}$, where $W_j^x \in \mathbb{R}^{r \times p}, W_j^z \in \mathbb{R}^{r \times q}$.

Now observe that in each level of the network, we are in effect performing a 1D convolution over the hidden units $z_{1:T}^{(i)}$. The output of this convolution is then passed through the activation function $f$. Formally, with $W \in \mathbb{R}^{r \times q}$ as the kernel weight matrix, the computation in layer $i$ can be summarized as follows (Figure 1b):

$$\hat{z}_{1:T}^{(i+1)} = \text{Conv1D}\left(z_{1:T}^{(i)}; W\right) + \tilde{x}_{1:T}, \qquad z_{1:T}^{(i+1)} = f\left(\hat{z}_{1:T}^{(i+1)}, z_{1:T-1}^{(i)}\right). \tag{3}$$

The resulting network operates in feed-forward fashion, with deeper elements having progressively larger receptive fields. There are, however, important differences from typical (temporal) convolutional networks. Notably, the filter matrix is shared across all layers. That is, the weights are tied not only across time but also across depth. (Vogel & Pock (2017) have previously tied weights across depth in image processing.) Another difference is that the transformed input sequence $\tilde{x}_{1:T}$ is directly injected into each hidden layer. These differences and their importance will be analyzed further in Section 4.

The activation function $f$ in Equation (3) can be any nonlinearity that processes the pre-activation output $\hat{z}_{1:T}^{(i+1)}$ and the output from the previous layer $z_{1:T-1}^{(i)}$. We will later describe an activation function based on the LSTM cell. The rationale for its use will become clearer in light of the analysis presented in the next section.

## 4 TRELLISNET, TCN, AND RNN

In this section, we analyze the relationships between trellis networks, convolutional networks, and recurrent networks. In particular, we show that trellis networks can serve as a bridge between convolutional and recurrent networks. On the one hand, TrellisNet is a special form of temporal convolutional networks (TCN); this has already been clear in Section 3 and will be discussed further in Section 4.1. On the other hand, any truncated RNN can be represented as a TrellisNet with special structure in the interlayer transformations; this will be the subject of Section 4.2. These connections allow TrellisNet to harness architectural elements and regularization techniques from both TCNs and RNNs; this will be summarized in Section 4.3.

### 4.1 TRELLISNET AND TCN

We briefly introduce TCNs here, and refer the readers to Bai et al. (2018) for a more thorough discussion. Briefly, a temporal convolutional network (TCN) is a ConvNet that uses one-dimensional convolutions over the sequence. The convolutions are *causal*, meaning that, at each layer, the transformation at time $t$ can only depend on previous layer units at times $t$ or earlier, not from later points in time. Such approaches were used going back to the late 1980s, under the name of "time-delay neural networks" (Waibel et al., 1989), and have received significant interest in recent years due to their application in architectures such as WaveNet (van den Oord et al., 2016).

In essence, TrellisNet is a special kind of temporal convolutional network. TCNs have two distinctive characteristics: 1) causal convolution in each layer to satisfy the causality constraint and 2) deep stacking of layers to increase the effective history length (i.e. receptive field). Trellis networks have both of these characteristics. The basic model presented in Section 3 can easily be elaborated with larger kernel sizes, dilated convolutions, and other architectural elements used in TCNs; some of these are reviewed further in Section 4.3.

However, TrellisNet is not a general TCN. As mentioned in Section 3, two important differences are: 1) the weights are tied across layers and 2) the linearly transformed input $\tilde{x}_{1:T}$ is injected into each layer. Weight tying can be viewed as a form of regularization that can stabilize training, support generalization, and significantly reduce the size of the model. Input injection mixes deep features with the original sequence. These structural characteristics will be further illuminated by the connection between trellis networks and recurrent networks, presented next.

### 4.2 TRELLISNET AND RNN

Recurrent networks appear fundamentally different from convolutional networks. Instead of operating on all elements of a sequence in parallel in each layer, an RNN processes one input element at a time and unrolls in the time dimension. Given a non-linearity $g$ (which could be a sigmoid or a more elaborate cell), we can summarize the transformations in an $L$-layer RNN at time-step $t$ as follows:

$$h_t^{(i)} = g\left(W_{hx}^{(i)} h_t^{(i-1)} + W_{hh}^{(i)} h_{t-1}^{(i)}\right) \quad \text{for } 1 \le i \le L, \qquad h_t^{(0)} = x_t. \tag{4}$$

Despite the apparent differences, we will now show that any RNN unrolled to a finite length is equivalent to a TrellisNet with special sparsity structure in the kernel matrix $W$. We begin by formally defining the notion of a truncated (i.e. finite-horizon) RNN.

**Definition 1.** *Given an RNN $\rho$, a corresponding **M-truncated RNN** $\rho^M$, applied to the sequence $x_{1:T}$, produces at time step $t$ the output $y_t$ by applying $\rho$ to the sequence $x_{t-M+1:t}$ (here $x_{<0} = 0$).*

**Theorem 1.** *Let $\rho^M$ be an $M$-truncated RNN with $L$ layers and hidden unit dimensionality $d$. Then there exists an equivalent TrellisNet $\tau$ with depth $(M + L - 1)$ and layer width (i.e. number of channels in each hidden layer) $Ld$. Specifically, for any $x_{1:T}$, $\rho^M(x_{1:T}) = \tau_{L(d-1)+1:Ld}(x_{1:T})$ (i.e. the TrellisNet outputs contain the RNN outputs).*

Theorem 1 states that any $M$-truncated RNN can be represented as a TrellisNet. How severe of a restriction is $M$-truncation? Note that $M$-truncation is intimately related to truncated backpropagation-through-time (BPTT), used pervasively in training recurrent networks on long sequences. While RNNs can in principle retain unlimited history, there is both empirical and theoretical evidence that the memory horizon of RNNs is bounded (Bai et al., 2018; Khandelwal et al., 2018;

Miller & Hardt, 2018). Furthermore, if desired, TrellisNets can recover exactly a common method of applying RNNs to long sequences – hidden state repackaging, i.e. copying the hidden state across subsequences. This is accomplished using an analogous form of hidden state repackaging, detailed in Appendix B.

*Proof of Theorem 1.* Let $h_{t,t'}^{(i)} \in \mathbb{R}^d$ be the hidden state at time $t$ and layer $i$ of the truncated RNN $\rho^{t-t'+1}$ (i.e., the RNN begun at time $t'$ and run until time $t$). Note that without truncation, history starts at time $t' = 1$, so the hidden state $h_t^{(i)}$ of $\rho$ can be equivalently expressed as $h_{t,1}^{(i)}$. When $t' > t$, we define $h_{t,t'} = 0$ (i.e. no history information if the clock starts in the future).

By assumption, $\rho^M$ is an RNN defined by the following parameters: $\{W_{hx}^{(i)}, W_{hh}^{(i)}, g, M\}$, where $W_{hh}^{(i)} \in \mathbb{R}^{w \times d}$ for all $i$, $W_{hx}^{(1)} \in \mathbb{R}^{w \times p}$, and $W_{hx}^{(i)} \in \mathbb{R}^{w \times d}$ for all $i = 2, \dots, L$ are the weight matrices at each layer ($w$ is the dimension of pre-activation output). We now construct a TrellisNet $\tau$ according to the exact definition in Section 3, with parameters $\{W_1, W_2, f\}$, where

$$W_1 = \begin{bmatrix} 0 & W_{hh}^{(1)} & 0 & \dots & 0 \\ 0 & 0 & W_{hh}^{(2)} & \dots & 0 \\ \vdots & \vdots & \vdots & \ddots & \vdots \\ 0 & 0 & 0 & \dots & W_{hh}^{(L)} \end{bmatrix}, W_2 = \begin{bmatrix} W_{hx}^{(1)} & 0 & \dots & 0 & 0 \\ 0 & W_{hx}^{(2)} & \dots & 0 & 0 \\ \vdots & \vdots & \ddots & \vdots & \vdots \\ 0 & 0 & \dots & W_{hx}^{(L)} & 0 \end{bmatrix}, \tag{5}$$

such that $W_1, W_2 \in \mathbb{R}^{Lw \times (p+Ld)}$. We define a nonlinearity $f$ by $f(\alpha, \beta) = g(\alpha)$ (i.e. applying $g$ only on the first entry).

Let $t \in [T]$, $j \geq 0$ be arbitrary and fixed. We now claim that the hidden unit at time $t$ and layer $j$ of TrellisNet $\tau$ can be expressed in terms of hidden units at time $t$ in truncated forms of $\rho$:

$$z_t^{(j)} = \begin{bmatrix} h_{t,t-j+1}^{(1)} & h_{t,t-j+2}^{(2)} & \dots & h_{t,t-j+L}^{(L)} \end{bmatrix}^\top \in \mathbb{R}^{Ld}, \tag{6}$$

where $z_t^{(j)}$ is the time-$t$ hidden state at layer $j$ of $\tau$ and $h_{t,t'}^{(i)}$ is the time-$t$ hidden state at layer $i$ of $\rho^{t-t'+1}$.

We prove Eq. (6) by induction on $j$. As a base case, consider $j = 0$; i.e. the input layer of $\tau$. Since $h_{t,t'} = 0$ when $t' > t$, we have that $z_j^{(0)} = [0 \ 0 \ \dots \ 0]^\top$. (Recall that in the input layer of TrellisNet we initialize $z_t^{(0)} = \mathbf{0}$.) For the inductive step, suppose Eq. (6) holds for layer $j$, and consider layer $j + 1$. By the feed-forward transformation of TrellisNet defined in Eq. (2) and the nonlinearity $f$ we defined above, we have:

$$\hat{z}_t^{(j+1)} = W_1 \begin{bmatrix} x_{t-1} \\ z_{t-1}^{(j)} \end{bmatrix} + W_2 \begin{bmatrix} x_t \\ z_t^{(j)} \end{bmatrix} \tag{7}$$

$$= \begin{bmatrix} 0 & W_{hh}^{(1)} & 0 & \dots & 0 \\ 0 & 0 & W_{hh}^{(2)} & \dots & 0 \\ \vdots & \vdots & \vdots & \ddots & \vdots \\ 0 & 0 & 0 & \dots & W_{hh}^{(L)} \end{bmatrix} \begin{bmatrix} x_{t-1} \\ h_{t-1,t-j}^{(1)} \\ \vdots \\ h_{t-1,t-j+L-1}^{(L)} \end{bmatrix} + \begin{bmatrix} W_{hx}^{(1)} & 0 & \dots & 0 & 0 \\ 0 & W_{hx}^{(2)} & \dots & 0 & 0 \\ \vdots & \vdots & \ddots & \vdots & \vdots \\ 0 & 0 & \dots & W_{hx}^{(L)} & 0 \end{bmatrix} \begin{bmatrix} x_t \\ h_{t,t-j+1}^{(1)} \\ \vdots \\ h_{t,t-j+L}^{(L)} \end{bmatrix} \tag{8}$$

$$= \begin{bmatrix} W_{hh}^{(1)} h_{t-1,t-j}^{(1)} + W_{hx}^{(1)} x_t \\ \vdots \\ W_{hh}^{(L)} h_{t-1,t-j+L-1}^{(L)} + W_{hx}^{(L)} h_{t,t-j+L-1}^{(L-1)} \end{bmatrix} \tag{9}$$

$$z_t^{(j+1)} = f(\hat{z}_t^{(j+1)}, z_{t-1}^{(j)}) = g(\hat{z}_t^{(j+1)}) = \begin{bmatrix} h_{t,t-j}^{(1)} & h_{t,t-j+1}^{(2)} & \dots & h_{t,t-j+L-1}^{(L)} \end{bmatrix}^\top \tag{10}$$

where in Eq. (10) we apply the RNN non-linearity $g$ following Eq. (4). Therefore, by induction, we have shown that Eq. (6) holds for all $j \geq 0$.

If TrellisNet $\tau$ has $M+L-1$ layers, then at the final layer we have $z_t^{(M+L-1)} = [\dots \ \dots \ h_{t,t+1-M}^{(L)}]^\top$. Since $\rho^M$ is an $L$-layer $M$-truncated RNN, this (taking the last $d$ channels of $z_t^{(M+L-1)}$) is exactly the output of $\rho^M$ at time $t$.

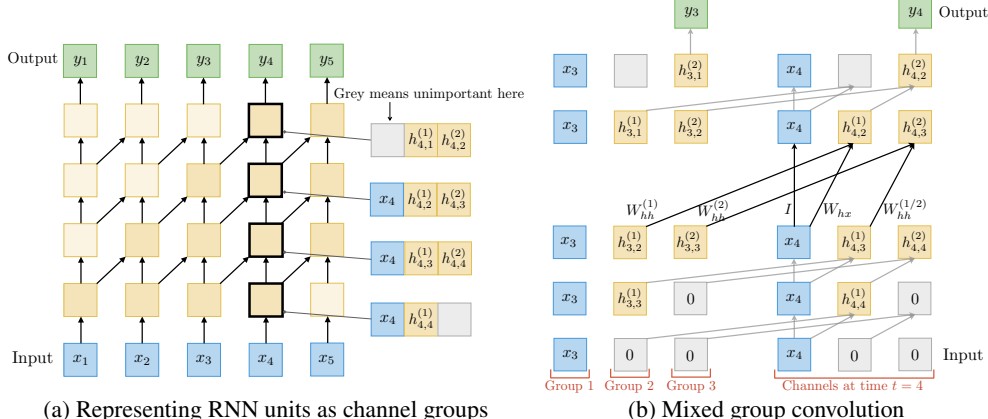

(a) Representing RNN units as channel groups

(b) Mixed group convolution

Figure 2: Representing a truncated 2-layer RNN $\rho^M$ as a trellis network $\tau$. **(a)** Each unit of $\tau$ has three groups, which house the input, first-layer hidden vector, and second-layer hidden vector of $\rho^M$, respectively. **(b)** Each group in the hidden unit of $\tau$ in level $i+1$ at time step $t+1$ is computed by a linear combination of appropriate groups of hidden units in level $i$ at time steps $t$ and $t+1$. The linear transformations form a mixed group convolution that reproduces computation in $\rho^M$. (Nonlinearities not shown for clarity.)

In other words, we have shown that $\rho^M$ is equivalent to a TrellisNet with sparse kernel matrices $W_1, W_2$. This completes the proof. ☐

Note that the convolutions in the TrellisNet $\tau$ constructed in Theorem 1 are sparse, as shown in Eq. (5). They are related to group convolutions (Krizhevsky et al., 2012), but have an unusual form because group $k$ at time $t$ is convolved with group $k-1$ at time $t+1$. We refer to these as mixed group convolutions. Moreover, while Theorem 1 assumes that all layers of $\rho^M$ have the same dimensionality $d$ for clarity, the proof easily generalizes to cases where each layer has different widths.

For didactic purposes, we recap and illustrate the construction in the case of a 2-layer RNN. The key challenge is that a naïve unrolling of the RNN into a feed-forward network does not produce a convolutional network, since the linear transformation weights are not constant across a layer. The solution, illustrated in Figure 2a, is to organize each hidden unit into groups of channels, such that each TrellisNet unit represents 3 RNN units simultaneously (for $x_t, h_t^{(1)}, h_t^{(2)}$). Each TrellisNet unit thus has $(p + 2d)$ channels. The interlayer transformation can then be expressed as a mixed group convolution, illustrated in Figure 2b. This can be represented as a sparse convolution with the structure given in Eq. (5) (with $L = 2$). Applying the nonlinearity $g$ on the pre-activation output, this exactly reproduces the transformations in the original 2-layer RNN.

The TrellisNet that emerges from this construction has special sparsity structure in the weight matrix. It stands to reason that a general TrellisNet with an unconstrained (dense) weight matrix $W$ may have greater expressive power: it can model a broader class of transformations than the original RNN $\rho^M$. Note that while the hidden channels of the TrellisNet $\tau$ constructed in the proof of Theorem 1 are naturally arranged into groups that represent different layers of the RNN $\rho^M$ (Eq. (6)), an unconstrained dense weight matrix $W$ no longer admits such an interpretation. A model defined by a dense weight matrix is fundamentally distinct from the RNN $\rho^M$ that served as our point of departure. We take advantage of this expressivity and use general weight matrices $W$, as presented in Section 3, in our experiments. Our ablation analysis will show that such generalized dense transformations are beneficial, even when model capacity is controlled for.

The proof of Theorem 1 did not delve into the inner structure of the nonlinear transformation $g$ in RNN (or $f$ in the constructed TrellisNet). For a vanilla RNN, for instance, $f$ is usually an elementwise sigmoid or $\tanh$ function. But the construction in Theorem 1 applies just as well to RNNs with structured cells, such as LSTMs and GRUs. We adopt LSTM cells for the TrellisNets in our experiments and provide a detailed treatment of this nonlinearity in Section 5.1 and Appendix A.

### 4.3 TRELLISNET AS A BRIDGE BETWEEN RECURRENT AND CONVOLUTIONAL MODELS

In Section 4.1 we concluded that TrellisNet is a special kind of TCN, characterized by weight tying and input injection. In Section 4.2 we established that TrellisNet is a generalization of truncated

RNNs. These connections along with the construction in our proof of Theorem 1 allow TrellisNets to benefit significantly from techniques developed originally for RNNs, while also incorporating architectural and algorithmic motifs developed for convolutional networks. We summarize a number of techniques here. From recurrent networks, we can integrate 1) structured nonlinear activations (e.g. LSTM and GRU gates); 2) variational RNN dropout (Gal & Ghahramani, 2016); 3) recurrent DropConnect (Merity et al., 2018b); and 4) history compression and repackaging. From convolutional networks, we can adapt 1) larger kernels and dilated convolutions (Yu & Koltun, 2016); 2) auxiliary losses at intermediate layers (Lee et al., 2015; Xie & Tu, 2015); 3) weight normalization (Salimans & Kingma, 2016); and 4) parallel convolutional processing. Being able to directly incorporate techniques from both streams of research is one of the benefits of trellis networks. We leverage this in our experiments and provide a more comprehensive treatment of these adaptations in Appendix B.

## 5 EXPERIMENTS

### 5.1 A TRELLISNET WITH GATED ACTIVATION

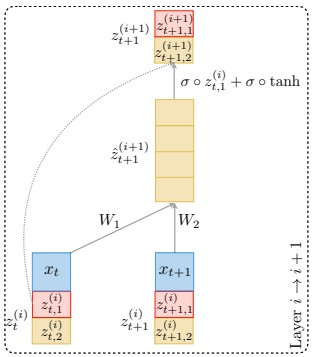

Figure 3: A gated activation based on the LSTM cell.

In our description of generic trellis networks in Section 3, the activation function $f$ can be any nonlinearity that computes $z_{1:T}^{(i+1)}$ based on $\hat{z}_{1:T}^{(i+1)}$ and $z_{1:T-1}^{(i)}$. In experiments, we use a gated activation based on the LSTM cell. Gated activations have been used before in convolutional networks for sequence modeling (van den Oord et al., 2016; Dauphin et al., 2017). Our choice is inspired directly by Theorem 1, which suggests incorporating an existing RNN cell into TrellisNet. We use the LSTM cell due to its effectiveness in recurrent networks (Jozefowicz et al., 2015; Greff et al., 2017; Melis et al., 2018). We summarize the construction here; a more detailed treatment can be found in Appendix A.

In an LSTM cell, three information-controlling gates are computed at time $t$. Moreover, there is a cell state that does not participate in the hidden-to-hidden transformations but is updated in every step using the result from the gated activations. We integrate the LSTM cell into the TrellisNet as follows (Figure 3):

$$\hat{z}_{t+1}^{(i+1)} = W_1 \begin{bmatrix} x_t \\ z_{t,2}^{(i)} \end{bmatrix} + W_2 \begin{bmatrix} x_{t+1} \\ z_{t+1,2}^{(i)} \end{bmatrix} = \begin{bmatrix} \hat{z}_{t+1,1} & \hat{z}_{t+1,2} & \hat{z}_{t+1,3} & \hat{z}_{t+1,4} \end{bmatrix}^\top \quad (11)$$

$$\begin{aligned} z_{t+1,1}^{(i+1)} &= \sigma(\hat{z}_{t+1,1}) \circ z_{t,1}^{(i)} + \sigma(\hat{z}_{t+1,2}) \circ \tanh(\hat{z}_{t+1,3}) \\ z_{t+1,2}^{(i+1)} &= \sigma(\hat{z}_{t+1,4}) \circ \tanh(z_{t+1,1}^{(i+1)}) \end{aligned} \quad (12; \text{Gated activation } f)$$

Thus the linear transformation in each layer of the TrellisNet produces a pre-activation feature $\hat{z}_{t+1}$ with $r = 4q$ feature channels, which are then processed by elementwise transformations and Hadamard products to yield the final output $z_{t+1}^{(i+1)} = \left( z_{t+1,1}^{(i+1)}, z_{t+1,2}^{(i+1)} \right)$ of the layer.

### 5.2 RESULTS

We evaluate trellis networks on word-level and character-level language modeling on the standard Penn Treebank (PTB) dataset (Marcus et al., 1993; Mikolov et al., 2010), large-scale word-level modeling on WikiText-103 (WT103) (Merity et al., 2017), and standard stress tests used to study long-range information propagation in sequence models: sequential MNIST, permuted MNIST (PMNIST), and sequential CIFAR-10 (Chang et al., 2017; Bai et al., 2018; Trinh et al., 2018). Note that these tasks are on very different scales, with unique properties that challenge sequence models in different ways. For example, word-level PTB is a small dataset that a typical model easily overfits, so judicious regularization is essential. WT103 is a hundred times larger, with less danger of overfitting, but with a vocabulary size of 268K that makes training more challenging (and precludes the application of techniques such as mixture of softmaxes (Yang et al., 2018)). A more complete description of these tasks and their characteristics can be found in Appendix C.

Table 1: Test perplexities (ppl) on word-level language modeling with the PTB corpus. $^\ell$ means lower is better.

| Word-level Penn Treebank (PTB) | | |
|---|---|---|
| Model | Size | Test perplexity$^\ell$ |
| Generic TCN (Bai et al., 2018) | 13M | 88.68 |
| Variational LSTM (Gal & Ghahramani, 2016) | 66M | 73.4 |
| NAS Cell (Zoph & Le, 2017) | 54M | 62.4 |
| AWD-LSTM (Merity et al., 2018b) | 24M | 58.8 |
| (Black-box tuned) NAS (Melis et al., 2018) | 24M | 59.7 |
| (Black-box tuned) LSTM + skip conn. (Melis et al., 2018) | 24M | 58.3 |
| AWD-LSTM-MoC (Yang et al., 2018) | 22M | 57.55 |
| DARTS (Liu et al., 2018) | 23M | 56.10 |
| AWD-LSTM-MoS (Yang et al., 2018) | 24M | 55.97 |
| ENAS (Pham et al., 2018) | 24M | 55.80 |
| Ours - TrellisNet | 24M | 56.97 |
| Ours - TrellisNet (1.4x larger) | 33M | 56.80 |
| Ours - TrellisNet-MoS | 25M | 54.67 |
| **Ours - TrellisNet-MoS (1.4x larger)** | 34M | **54.19** |

Table 2: Test perplexities (ppl) on word-level language modeling with the WT103 corpus.

| Word-level WikiText-103 (WT103) | | |
|---|---|---|
| Model | Size | Test perplexity$^\ell$ |
| LSTM (Grave et al., 2017b) | - | 48.7 |
| LSTM+continuous cache (Grave et al., 2017b) | - | 40.8 |
| Generic TCN (Bai et al., 2018) | 150M | 45.2 |
| Gated Linear ConvNet (Dauphin et al., 2017) | 230M | 37.2 |
| AWD-QRNN (Merity et al., 2018a) | 159M | 33.0 |
| Relational Memory Core (Santoro et al., 2018) | 195M | 31.6 |
| **Ours - TrellisNet** | 180M | **29.19** |

The prior state of the art on these tasks was set by completely different models, such as AWD-LSTM on character-level PTB (Merity et al., 2018a), neural architecture search on word-level PTB (Pham et al., 2018), and the self-attention-based Relational Memory Core on WikiText-103 (Santoro et al., 2018). We use trellis networks on all tasks and outperform the respective state-of-the-art models on each. For example, on word-level Penn Treebank, TrellisNet outperforms by a good margin the recent results of Melis et al. (2018), which used the Google Vizier service for exhaustive hyperparameter tuning, as well as the recent neural architecture search work of Pham et al. (2018). On WikiText-103, a trellis network outperforms by 7.6% the Relational Memory Core (Santoro et al., 2018) and by 11.5% the thorough optimization work of Merity et al. (2018a).

Many hyperparameters we use are adapted directly from prior work on recurrent networks. (As highlighted in Section 4.3, many techniques can be carried over directly from RNNs.) For others, we perform a basic grid search. We decay the learning rate by a fixed factor once validation error plateaus. All hyperparameters are reported in Appendix D, along with an ablation study.

**Word-level language modeling.** For word-level language modeling, we use PTB and WT103. The results on PTB are listed in Table 1. TrellisNet sets a new state of the art on PTB, both with and without mixture of softmaxes (Yang et al., 2018), outperforming all previously published results by more than one unit of perplexity.

WT103 is 110 times larger than PTB, with vocabulary size 268K. We follow prior work and use the adaptive softmax (Grave et al., 2017a), which improves memory efficiency by assigning higher capacity to more frequent words. The results are listed in Table 2. TrellisNet sets a new state of the art on this dataset as well, with perplexity 29.19: about 7.6% better than the contemporaneous

Table 3: Test bits-per-character (bpc) on character-level language modeling with the PTB corpus.

| Char-level PTB | | |
|---|---|---|
| Model | Size | Test bpc$^{\ell}$ |
| Generic TCN (Bai et al., 2018) | 3.0M | 1.31 |
| Independently RNN (Li et al., 2018) | 12.0M | 1.23 |
| Hyper LSTM (Ha et al., 2017) | 14.4M | 1.219 |
| NAS Cell (Zoph & Le, 2017) | 16.3M | 1.214 |
| Fast-Slow-LSTM-2 (Mujika et al., 2017) | 7.2M | 1.19 |
| Quasi-RNN (Merity et al., 2018a) | 13.8M | 1.187 |
| AWD-LSTM (Merity et al., 2018a) | 13.8M | 1.175 |
| **Ours - TrellisNet** | 13.4M | **1.158** |

Table 4: Test accuracies on long-range modeling benchmarks. $^{h}$ means higher is better.

| Model | Seq. MNIST Test acc.$^{h}$ | Permuted MNIST Test acc.$^{h}$ | Seq. CIFAR-10 Test acc.$^{h}$ |
|---|---|---|---|
| Dilated GRU (Chang et al., 2017) | 99.0 | 94.6 | - |
| IndRNN (Li et al., 2018) | 99.0 | 96.0 | - |
| Generic TCN (Bai et al., 2018) | 99.0 | 97.2 | - |
| $r$-LSTM w/ Aux. Loss (Trinh et al., 2018) | 98.4 | 95.2 | 72.2 |
| Transformer (self-attention) (Trinh et al., 2018) | 98.9 | 97.9 | 62.2 |
| **Ours - TrellisNet** | **99.20** | **98.13** | **73.42** |

self-attention-based Relational Memory Core (RMC) (Santoro et al., 2018). TrellisNet achieves this better accuracy with much faster convergence: 25 epochs, versus 90 for RMC.

**Character-level language modeling.** When used for character-level modeling, PTB is a medium-scale dataset with stronger long-term dependencies between characters. We thus use a deeper network as well as techniques such as weight normalization (Salimans & Kingma, 2016) and deep supervision (Lee et al., 2015; Xie & Tu, 2015). The results are listed in Table 3. TrellisNet sets a new state of the art with 1.158 bpc, outperforming the recent results of Merity et al. (2018a) by a comfortable margin.

**Long-range modeling with Sequential MNIST, PMNIST, and CIFAR-10.** We also evaluate the TrellisNet for ability to model long-term dependencies. In the Sequential MNIST, PMNIST, and CIFAR-10 tasks, images are processed as long sequences, one pixel at a time (Chang et al., 2017; Bai et al., 2018; Trinh et al., 2018). Our model has 8M parameters, in alignment with prior work. To cover the larger context, we use dilated convolutions in intermediate layers, adopting a common architectural element from TCNs (Yu & Koltun, 2016; van den Oord et al., 2016; Bai et al., 2018). The results are listed in Table 4. Note that the performance of prior models is inconsistent. The Transformer works well on MNIST but fairs poorly on CIFAR-10, while $r$-LSTM with unsupervised auxiliary losses achieves good results on CIFAR-10 but underperforms on Permuted MNIST. TrellisNet outperforms all these models on all three tasks.

## 6  DISCUSSION

We presented trellis networks, a new architecture for sequence modeling. Trellis networks form a structural bridge between convolutional and recurrent models. This enables direct assimilation of many techniques designed for either of these two architectural families. We leverage these connections to train high-performing trellis networks that set a new state of the art on highly competitive language modeling benchmarks. Beyond the empirical gains, we hope that trellis networks will serve as a step towards deeper and more unified understanding of sequence modeling.

There are many exciting opportunities for future work. First, we have not conducted thorough performance optimizations on trellis networks. For example, architecture search on the structure of the gated activation $f$ may yield a higher-performing activation function than the classic LSTM cell

we used (Zoph & Le, 2017; Pham et al., 2018). Likewise, principled hyperparameter tuning will likely improve modeling accuracy beyond the levels we have observed (Melis et al., 2018). Future work can also explore acceleration schemes that speed up training and inference.

Another significant opportunity is to establish connections between trellis networks and self-attention-based architectures (Transformers) (Vaswani et al., 2017; Santoro et al., 2018; Chen et al., 2018), thus unifying all three major contemporary approaches to sequence modeling. Finally, we look forward to seeing applications of trellis networks to industrial-scale challenges such as machine translation.

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

## A  EXPRESSING AN LSTM AS A TRELLISNET

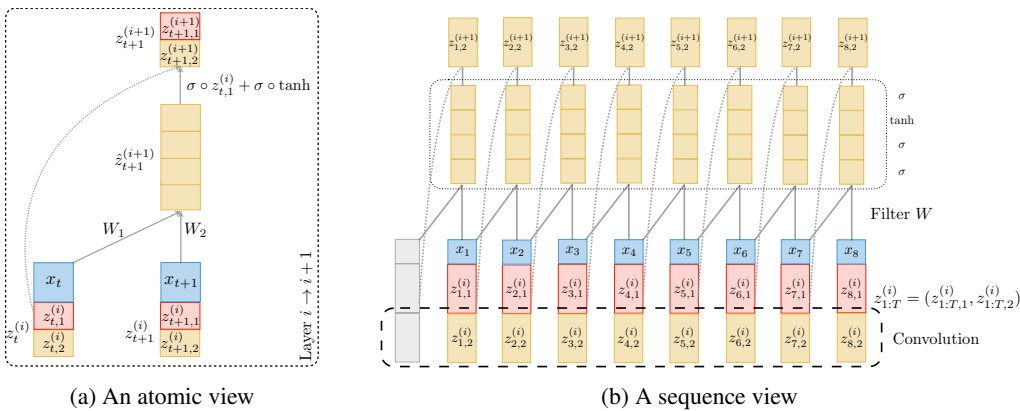

(a) An atomic view  (b) A sequence view

Figure 4: A TrellisNet with an LSTM nonlinearity, at an atomic level and on a longer sequence.

Here we trace in more detail the transformation of an LSTM into a TrellisNet. This is an application of Theorem 1. The nonlinear activation has been examined in Section 5.1. We will walk through the construction again here.

In each time step, an LSTM cell computes the following:

$$
\begin{aligned}
f_t^{(\ell)} &= \sigma(W_f h_t^{(\ell-1)} + U_f h_{t-1}^{(\ell)}) \quad i_t^{(\ell)} = \sigma(W_i h_t^{(\ell-1)} + U_i h_{t-1}^{(\ell)}) \quad g_t^{(\ell)} = \tanh(W_g h_t^{(\ell-1)} + U_g h_{t-1}^{(\ell)}) \\
o_t^{(\ell)} &= \sigma(W_o h_t^{(\ell-1)} + U_o h_{t-1}^{(\ell)}) \quad c_t^{(\ell)} = f_t^{(\ell)} \circ c_{t-1}^{(\ell)} + i_t^{(\ell)} \circ g_t^{(\ell)} \quad h_t^{(\ell)} = o_t^{(\ell)} \circ \tanh(c_t^{(\ell)})
\end{aligned}
\tag{13}
$$

where $h_t^{(0)} = x_t$, and $f_t, i_t, o_t$ are typically called the *forget*, *input*, and *output* gates. By a similar construction to how we defined $\tau$ in Theorem 1, to recover an LSTM the mixed group convolution needs to produce $3q$ more channels for these gated outputs, which have the form $f_{t,t'}, i_{t,t'}$ and $g_{t,t'}$ (see Figure 5 for an example). In addition, at each layer of the mixed group convolution, the network also needs to maintain a group of channels for cell states $c_{t,t'}$. Note that in an LSTM network, $c_t$ is updated "synchronously" with $h_t$, so we can similarly write

$$
c_{t,t'}^{(1)} = f_{t,t'}^{(1)} \circ c_{t-1,t'}^{(1)} + i_{t,t'}^{(1)} \circ g_{t,t'}^{(1)} \qquad h_{t,t'}^{(1)} = o_{t,t'}^{(1)} \circ \tanh(c_{t,t'}^{(1)})
\tag{14}
$$

Based on these changes, we show in Figure 4 an atomic and a sequence view of TrellisNet with the LSTM activation. The hidden units $z_{1:T}$ consist of two parts: $z_{1:T,1}$, which gets updated directly via the gated activations (akin to LSTM cell states), and $z_{1:T,2}$, which is processed by parameterized convolutions (akin to LSTM hidden states). Formally, in layer $i$:

$$
\begin{aligned}
\hat{z}_{1:T}^{(i+1)} &= \text{Conv1D}(z_{1:T,2}^{(i)}; W) + \tilde{x}_{1:T} = [\hat{z}_{1:T,1} \quad \hat{z}_{1:T,2} \quad \hat{z}_{1:T,3} \quad \hat{z}_{1:T,4}]^\top \\
z_{1:T,1}^{(i+1)} &= \sigma(\hat{z}_{1:T,1}) \circ z_{0:T-1,1}^{(i)} + \sigma(\hat{z}_{1:T,2}) \circ \tanh(\hat{z}_{1:T,3}) \\
z_{1:T,2}^{(i+1)} &= \sigma(\hat{z}_{1:T,4}) \circ \tanh(z_{1:T,1}^{(i+1)})
\end{aligned}
$$

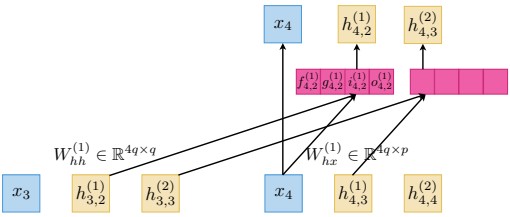

Figure 5: A 2-layer LSTM is expressed as a trellis network with mixed group convolutions on four groups of feature channels. (Partial view.)

# B   OPTIMIZING AND REGULARIZING TRELLISNET WITH RNN AND TCN METHODS

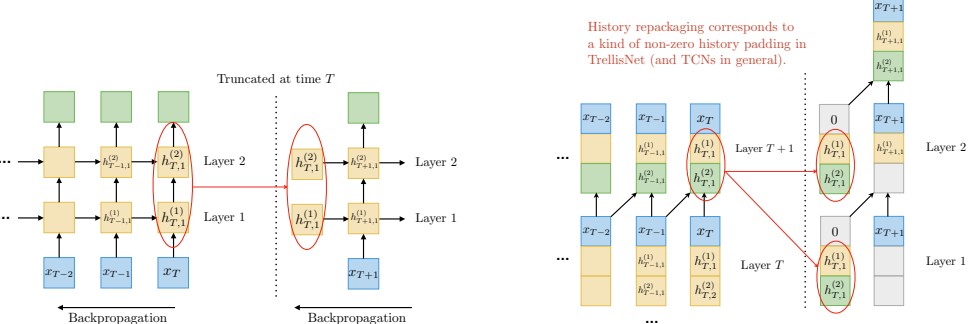

(a) History repackaging between truncated sequences in recurrent networks.

(b) History repackaging in mixed group convolutions, where we write out $z_t$ explicitly by Eq. (6).

Figure 6: Using the equivalence established by Theorem 1, we can transfer the notion of history repackaging in recurrent networks to trellis networks.

In Section 4, we formally described the relationship between TrellisNets, RNNs, and temporal convolutional networks (TCN). On the one hand, TrellisNet is a special TCN (with weight-tying and input injection), while on the other hand it can also express any structured RNN via a sparse convolutional kernel. These relationships open clear paths for applying techniques developed for either recurrent or convolutional networks. We summarize below some of the techniques that can be applied in this way to TrellisNet, categorizing them as either inspired by RNNs or TCNs.

## B.1   FROM RECURRENT NETWORKS

**History repackaging.** One theoretical advantage of RNNs is their ability to represent a history of infinite length. However, in many applications, sequence lengths are too long for infinite backpropagation during training. A typical solution is to partition the sequence into smaller subsequences and perform truncated backpropagation through time (BPTT) on each. At sequence boundaries, the hidden state $h_t$ is "repackaged" and passed onto the next RNN sequence. Thus gradient flow stops at sequence boundaries (see Figure 6a). Such repackaging is also sometimes used at test time.

We can now map this repackaging procedure to trellis networks. As shown in Figure 6, the notion of passing the compressed history vector $h_t$ in an RNN corresponds to specific non-zero padding in the mixed group convolution of the corresponding TrellisNet. The padding is simply the channels from the last step of the final layer applied on the previous sequence (see Figure 6b, where without the repackaging padding, at layer 2 we will have $h_{T+1,T+1}^{(1)}$ instead of $h_{T+1,1}^{(1)}$). We illustrate this in Figure 6b, where we have written out $z_t^{(i)}$ in TrellisNet explicitly in the form of $h_{t,t'}$ according to Eq. (6). This suggests that instead of storing all effective history in memory, we can compress history in a feed-forward network to extend its history as well. For a general TrellisNet that employs a dense kernel, similarly, we can pass the hidden channels of the last step of the final layer in the previous sequence as the "history" padding for the next TrellisNet sequence (this works in both training and testing).

**Gated activations.** In general, the structured gates in RNN cells can be translated to gated activations in temporal convolutions, as we did in Appendix A in the case of an LSTM. While in the experiments we adopted the LSTM gating, other activations (e.g. GRUs (Cho et al., 2014) or activations found via architecture search (Zoph & Le, 2017)) can also be applied in trellis networks via the equivalence established in Theorem 1.

**RNN variational dropout.** Variational dropout (VD) for RNNs (Gal & Ghahramani, 2016) is a useful regularization scheme that applies the same mask at every time step within a layer (see Figure 7a). A direct translation of this technique from RNN to the group temporal convolution implies that we need to create a different mask for each diagonal of the network (i.e. each history starting point), as well as for each group of the mixed group convolution. We propose an alternative (and extremely simple) dropout scheme for TrellisNet, which is inspired by VD in RNNs as well as Theo-

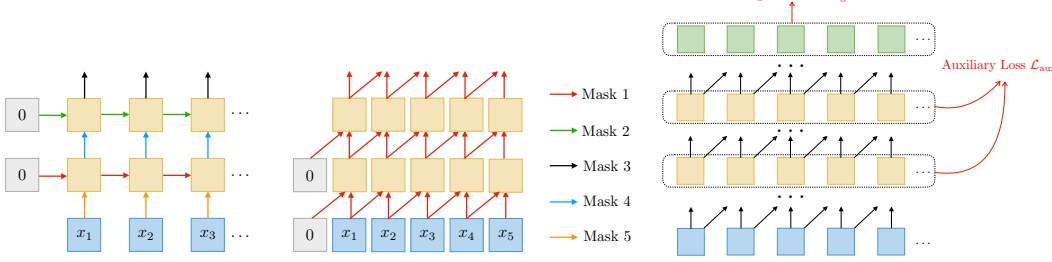

(a) Left: variational dropout (VD) in an RNN. Right: VD in a TrellisNet. Each color indicates a different dropout mask.

(b) Auxiliary loss on intermediate layers in a TrellisNet.

Figure 7: (a) RNN-inspired variational dropout. (b) ConvNet-inspired auxiliary losses.

rem 1. In each iteration, we apply the *same mask* on the post-activation outputs, at every time step in both the temporal dimension and depth dimension. That is, based on Eq. (6) in Theorem 1, we adapt VD to the TrellisNet setting by assuming $h_{t,t'\pm\delta} \approx h_{t,t'}$; see Figure 7a. Empirically, we found this dropout to work significantly better than other dropout schemes (e.g. drop certain channels entirely).

**Recurrent weight dropout/DropConnect.** We apply DropConnect on the TrellisNet kernel. Merity et al. (2018b) showed that regularizing hidden-to-hidden weights $W_{hh}$ can be useful in optimizing LSTM language models, and we carry this scheme over to trellis networks.

### B.2 FROM CONVOLUTIONAL NETWORKS

**Dense convolutional kernel.** Generalizing the convolution from a mixed group (sparse) convolution to a general (dense) one means the connections are no longer recurrent and we are computing directly on the hidden units with a large kernel, just like any temporal ConvNet.

**Deep supervision.** Recall that for sparse TrellisNet to recover truncated RNN, at each level the hidden units are of the form $h_{t,t'}$, representing the state at time $t$ *if we assume that history started at time $t'$* (Eq. (6)). We propose to inject the loss function at intermediate layers of the convolutional network (e.g. after every $\ell$ layers of transformations, where we call $\ell$ the auxiliary loss frequency). For example, during training, to predict an output at time $t$ with a $L$-layer TrellisNet, besides $z_t^{(L)}$ in the last layer, we can also apply the loss function on $z_t^{(L-\ell)}$, $z_t^{(L-2\ell)}$, etc. – where hidden units will predict with a shorter history because they are at lower levels of the network. This had been introduced for convolutional models in computer vision (Lee et al., 2015; Xie & Tu, 2015). The eventual loss of the network will be

$$\mathcal{L}_{\text{total}} = \mathcal{L}_{\text{orig}} + \lambda \cdot \mathcal{L}_{\text{aux}}, \tag{15}$$

where $\lambda$ is a fixed scaling factor that controls the weight of the auxiliary loss.

Note that this technique is not directly transferable (or applicable) to RNNs.

**Larger kernel and dilations (Yu & Koltun, 2016).** These techniques have been used in convolutional networks to more quickly increase the receptive field. They can be immediately applied to trellis networks. Note that the activation function $f$ of TrellisNet may need to change if we change the kernel size or dilation settings (e.g. with dilation $d$ and kernel size 2, the activation will be $f(\hat{z}_{1:T}^{(i)}, z_{1:T-d}^{(i)})$).

**Weight normalization (Salimans & Kingma, 2016).** Weight normalization (WN) is a technique that learns the direction and the magnitude of the weight matrix independently. Applying WN on the convolutional kernel was used in some prior works on temporal convolutional architectures (Dauphin et al., 2017; Bai et al., 2018), and have been found useful in regularizing the convolutional filters and boosting convergence.

**Parallelism.** Because TrellisNet is convolutional in nature, it can easily leverage the parallel processing in the convolution operation (which slides the kernel across the input features). We note that when the input sequence is relatively long, the predictions of the first few time steps will have insufficient history context compared to the predictions later in the sequence. This can be addressed by either history padding (mentioned in Appendix B.1) or chopping off the loss incurred by the first few time steps.

## C  BENCHMARK TASKS

**Word-level language modeling on Penn Treebank (PTB).** The original Penn Treebank (PTB) dataset selected 2,499 stories from a collection of almost 100K stories published in Wall Street Journal (WSJ) (Marcus et al., 1993). After Mikolov et al. (2010) processed the corpus, the PTB dataset contains 888K words for training, 70K for validation and 79K for testing, where each sentence is marked with an `<eos>` tag at its end. All of the numbers (e.g. in financial news) were replaced with a `?` symbol with many punctuations removed. Though small, PTB has been a highly studied dataset in the domain of language modeling (Miyamoto & Cho, 2016; Zilly et al., 2017; Merity et al., 2018b; Melis et al., 2018; Yang et al., 2018). Due to its relatively small size, many computational models can easily overfit on word-level PTB. Therefore, good regularization methods and optimization techniques designed for sequence models are especially important on this benchmark task (Merity et al., 2018b).

**Word-level language modeling on WikiText-103.** WikiText-103 (WT103) is 110 times larger than PTB, containing a training corpus from 28K lightly processed Wikipedia articles (Merity et al., 2017). In total, WT103 features a vocabulary size of about 268K[2], with 103M words for training, 218K words for validation, and 246K words for testing/evaluation. The WT103 corpus also retains the original case, punctuation and numbers in the raw data, all of which were removed from the PTB corpus. Moreover, since WT103 is composed of full articles (whereas PTB is sentence-based), it is better suited for testing long-term context retention. For these reasons, WT103 is typically considered much more representative and realistic than PTB (Merity et al., 2018a).

**Character-level language modeling on Penn Treebank (PTB).** When used for character-level language modeling, PTB is a medium size dataset that contains 5M chracters for training, 396K for validation, and 446K for testing, with an alphabet size of 50 (note: the `<eos>` tag that marks the end of a sentence in word-level tasks is now considered one character). While the alphabet size of char-level PTB is much smaller compared to the word-level vocabulary size (10K), there is much longer sequential token dependency because a sentence contains many more characters than words.

**Sequential and permuted MNIST classification.** The MNIST handwritten digits dataset (LeCun et al., 1989) contains 60K normalized training images and 10K testing images, all of size $28 \times 28$. In the sequential MNIST task, MNIST images are presented to the sequence model as a flattened $784 \times 1$ sequence for digit classification. Accurate predictions therefore require good long-term memory of the flattened pixels – longer than in most language modeling tasks. In the setting of permuted MNIST (PMNIST), the order of the sequence is permuted at random, so the network can no longer rely on local pixel features for classification.

**Sequential CIFAR-10 classification.** The CIFAR-10 dataset (Krizhevsky & Hinton, 2009) contains 50K images for training and 10K for testing, all of size $32 \times 32$. In the sequential CIFAR-10 task, these images are passed into the model one at each time step, flattended as in the MNIST tasks. Compared to sequential MNIST, this task is more challenging. For instance, CIFAR-10 contains more complex image structures and intra-class variations, and there are 3 channels to the input. Moreover, as the images are larger, a sequence model needs to have even longer memory than in sequential MNIST or PMNIST (Trinh et al., 2018).

## D  HYPERPARAMETERS AND ABLATION STUDY

Table 5 specifies the trellis networks used for the various tasks. There are a few things to note while reading the table. First, in training, we decay the learning rate once the validation error plateaus for a while (or according to some fixed schedule, such as after 100 epochs). Second, for auxiliary loss (see Appendix B for more details), we insert the loss function after every fixed number of layers in the network. This "frequency" is included below under the "Auxiliary Frequency" entry. Finally, the hidden dropout in the Table refers to the variational dropout we translated from RNNs (see Appendix B), which is applied at all hidden layers of the TrellisNet. Due to the insight from Theorem 1, many techniques in TrellisNet were translated directly from RNNs or TCNs. Thus, most of the hyperparameters were based on the numbers reported in prior works (e.g. embedding size, embedding dropout, hidden dropout, output dropout, optimizer, weight-decay, etc.) with minor

---

[2]As a reference, Oxford English Dictionary only contains less than 220K unique English words.

adjustments (Merity et al., 2018b; Yang et al., 2018; Bradbury et al., 2017; Merity et al., 2018a; Trinh et al., 2018; Bai et al., 2018; Santoro et al., 2018). For factors such as auxiliary loss weight and frequency, we perform a basic grid search.

Table 5: Models and hyperparameters used in experiments. "–" means not applicable/used.

| | Word-PTB (w/o MoS) | Word-PTB (w/ MoS) | Word-WT103 | Char-PTB | (P)MNIST/CIFAR-10 |
|---|---|---|---|---|---|
| Optimizer | SGD | SGD | Adam | Adam | Adam |
| Initial Learning Rate | 20 | 20 | 1e-3 | 2e-3 | 2e-3 |
| Hidden Size (i.e. $h_t$) | 1000 | 1000 | 2000 | 1000 | 100 |
| Output Size (only for MoS) | – | 480 | – | – | – |
| # of Experts (only for MoS) | – | 15 | – | – | – |
| Embedding Size | 400 | 280 | 512 | 200 | – |
| Embedding Dropout | 0.1 | 0.05 | 0.0 | 0.0 | – |
| Hidden (VD-based) Dropout | 0.28 | 0.28 | 0.1 | 0.3 | 0.2 |
| Output Dropout | 0.45 | 0.4 | 0.1 | 0.1 | 0.2 |
| Weight Dropout | 0.5 | 0.45 | 0.1 | 0.25 | 0.1 |
| # of Layers | 55 | 55 | 70 | 125 | 16 |
| Auxiliary Loss $\lambda$ | 0.05 | 0.05 | 0.08 | 0.3 | – |
| Auxiliary Frequency | 16 | 16 | 25 | 70 | – |
| Weight Normalization | – | – | ✓ | ✓ | ✓ |
| Gradient Clip | 0.225 | 0.2 | 0.1 | 0.2 | 0.5 |
| Weight Decay | 1e-6 | 1e-6 | 0.0 | 1e-6 | 1e-6 |
| **Model Size** | 24M | 25M | 180M | 13.4M | 8M |

We have also performed an ablation study on TrellisNet to study the influence of various ingredients and techniques on performance. The results are reported in Table 6. We conduct the study on word-level PTB using a TrellisNet with 24M parameters. When we study one factor (e.g. removing hidden dropout), all hyperparameters and settings remain the same as in column 1 of Table 5 (except for "Dense Kernel", where we adjust the number of hidden units so that the model size remains the same).

Table 6: Ablation study on word-level PTB (w/o MoS)

| | Model Size | Test ppl | Δ SOTA |
|---|---|---|---|
| TrellisNet | 24.1M | 56.97 | – |
| − Hidden (VD-based) Dropout | 24.1M | 64.69 | ↓ 7.72 |
| − Weight Dropout | 24.1M | 63.82 | ↓ 6.85 |
| − Auxiliary Losses | 24.1M | 57.99 | ↓ 1.02 |
| − Long Seq. Parallelism | 24.1M | 57.35 | ↓ 0.38 |
| − Dense Kernel (i.e. mixed group conv) | 24.1M | 59.18 | ↓ 2.21 |
| − Injected Input (every 2 layers instead) | 24.1M | 57.44 | ↓ 0.47 |
| − Injected Input (every 5 layers instead) | 24.1M | 59.75 | ↓ 2.78 |
| − Injected Input (every 10 layers instead) | 24.1M | 60.70 | ↓ 3.73 |
| − Injected Input (every 20 layers instead) | 24.1M | 74.91 | ↓ 17.94 |

