# OpenReview forum: "Trellis Networks for Sequence Modeling"
_ICLR.cc/2019/Conference_

### Official Review · AnonReviewer3 · 2018-11-02

**Rating:** 7
**Confidence:** 3

**Review:**

This paper introduces a novel architecture for sequence modeling, called the trellis network. The trellis network is in a sense a combination of RNNs and CNNs. The authors give a constructive proof that the trellis network is a special case of a truncated RNN. It also resembles CNNs since the neurons at higher levels have bigger receptive fields. As a result, techniques from RNN and CNN literature can be conveniently brought in and adapted to trellis network. The proposed method is evaluated on benchmark tasks and shows performance gain over existing methods.

The paper is well-written and easy to follow. The experimental study is extensive. The reviewer believes that this paper will potentially inspire future research along this direction. However, the novelty of the proposed method compared to the TCN seems limited: only weight sharing and input injection. It would be great to include the performance of the TCN on the PTB dataset, on both word and character levels in Table 1 and 2.

According to Theorem 1, to model an M-truncated L-layer RNN, a trellis network needs M + L − 1 layers. When M is large, it seems that a trellis network needs to be deep. Although this does not increase to model size due to weight sharing, does it significantly increase computation time, both during training and inference?

The review might have missed it, but what is the rationale behind the dotted link in Figure 1a, or the dependence of the activation function $f$ on $z_t^{(i)}$? It seems that it is neither motivated by RNNs nor CNNs. From RNN's point of view, as shown in the proof of Theorem 1, $f$ only depends on its first argument. From CNN's point of view, the model still gets the same reception field without using $z_t^{(i)}$.

Minor comments:
The authors might want to give the full name of TCN (temporal convolutional networks) and a short introduction in Section 2 or at the beginning of Section 4.

---

> ### Author Response · Authors · 2018-11-16
> **Response to AnonReviewer3**
>
> Thank you for the comments.
>
> Note that input injection and weight-tying across depth may seem reasonable in retrospect, but these ideas were not obvious a priori. They seemed quite alien to us until they emerged from the construction in Theorem 1.
>
> Regarding the performance of TCN on word- and char-level PTB, we have included these in the tables. They are much worse (by about 30 ppl on word-PTB and 0.14 bpc on char-PTB) than the prior SOTA results, despite the strong regularizations added to the original TCN. The significant improvement of TrellisNet over TCN is due to the ideas presented in our submission.
>
> Concerning the increase in computation time, the construction in Theorem 1 does produce deep networks with M + L − 1 layers. However, in practice we do not have to use this precise number. As highlighted in Section 4.3 and Appendix B, existing techniques can help us quickly expand the horizon (i.e. context size) of a TrellisNet, for example by larger kernel sizes, dilations, or history repackaging. We investigate the problem of long-range modeling in Section 5.2 (Table 4) as well, where the temporal dependency in sequential MNIST, permuted MNIST and sequential CIFAR-10 is typically over 700 or 1000. In that case, it would be impossible to fit a TrellisNet with (M+L-1) layers on a GPU. Our experiments have shown that with the help of dilations and other techniques, TrellisNets can achieve very strong performance with a smaller number of layers than a strict interpretation of Theorem 1 would suggest.
>
> Concerning the dotted link in Figure 1a, we would like to offer two kinds of perspectives, one from the RNN side and one from the TCN side. In the context of an RNN, this is quite similar to what an LSTM or a GRU cell would do, where the gating mechanism would involve the hidden state or cell state (e.g., c_{t-1} in LSTM) propagated from the previous time step (see Figure 3(a) and 4). From a TCN perspective, this resembles a residual connection, except that we shift the connection by one time step in the temporal dimension of the input tensor. An interesting connection is that the introduction of cell state propagation in LSTMs was used to alleviate the vanishing gradient problem, while residual connections have a similar effect and motivation in deep CNNs. The dotted connection in TrellisNet reflects both ideas.
>
> Concerning giving the full name of TCN and a brief introduction, we agree and have addressed this in the revision.

---

### Official Review · AnonReviewer1 · 2018-11-04
**Interesting formulation of a convolutional view of recurrent networks but the real impact of this model has yet to be shown**

**Rating:** 6
**Confidence:** 3

**Review:**

The authors propose a new type of neural network architecture for sequence modelling : Trellis Networks. A trellis network is a special case of temporal convolutional network with shared weights across time and layers and  with input at each layer. As stated by the authors, this architecture does not seem really interesting. The authors show that there exists  an equivalent Trellis Network to any truncated RNN and therefore that truncated RNN can be represented by temporal convolutional network. This result is not surprising since  truncated RNN can be  unrolled and that their time dependency is bounded.  The construction of the Trellis Network equivalent to a truncated RNN involves sparse weight matrices, therefore using full weight matrices provides a greater expressive power. One can regret that the authors do not explain what kind of modelling power one can gain with full weight matrices.

The author claim that bridging the gap between recurrent and convolutional neural networks with  Trellis Network allows to benefit from techniques form both kinds of networks. However, most of the techniques are already used with  convolutional networks.

Experiments are conducted with LSTM trellis network on several sequence modelling tasks : word-level and character-level language modelling, and sequence modelling in images (sequential MNIST, permuted MNIST  and sequential CIFAR-10). Trellis network yield very competitive results compare to recent state of the art models.

The ablation  study presented in Annex D Table 5 is interesting since it provides some hints on what is really useful in the model. It seems that full weight matrices are not the most interesting aspect (if dense kernel really concerns this aspect) and that the use if the input at every layer has most impact.

---

> ### Author Response · Authors · 2018-11-16
> **Response to AnonReviewer1**
>
> Thank you for the comments.
>
> First, while it may not seem surprising that a truncated RNN can be approximated by feed-forward networks in general (after all, any computational graph of finite length can be simply unrolled to form a feedforward network), we believe it is surprising that general RNNs can be represented with a simple kernel-2 TCN. The construction presented in our work has not been presented before and the details are quite interesting. For example, weight-tying across depth and input injection have not been used in TCNs for sequence modeling, and seemed quite strange to us until they emerged from the construction. These make sense in retrospect, but were not obvious a priori. Now that we have a better understanding of these architectural elements, they may see broader use in the community.
>
> Second, about the techniques from both kinds of networks, we included a list of examples that TrellisNet can absorb from TCNs/RNNs in section B.1. While methods such as dilations and deep supervision are more common in ConvNets, we found the RNN-inspired techniques are equally important for a well-performing TrellisNet. For instance, the variational dropout that was specifically designed for RNN sequence models, gated activations motivated by LSTM/GRU, and history repackaging are all rarely seen in ConvNets. Besides the ablation study, we have now included more results from the generic TCN in section 5 in our latest revision. In all cases, the TrellisNet (which benefits from the above modifications) vastly outperform the generic TCN, even though the TCN is equipped with standard ConvNet ideas (e.g., dilation). We do believe that the inspiration from RNNs contributes a lot to improving TrellisNet beyond the TCN boundary.
>
> Third, regarding the modeling power of the full weight matrices. We introduced mixed group convolutions (Figure 2) to model the “layers” in RNNs. Once we generalize to a full weight matrix, there is no longer an interpretation in terms of “layers”. This can be seen in Eq. (6) and Figure 2(b): using a full convolutional kernel essentially mixes hidden units with different starting histories at each layer of the TrellisNet; this is impossible in RNNs. Theoretically, the full weight matrices can learn the blocked diagonal structure of Eq. (5) by gradient updates, if such diagonal structure is truly the optimal arrangement for sequence task parameters. In other words, optimal RNNs can be recovered by TrellisNet through training. However, as we showed in large-scale tasks such as WT103, TrellisNet gains quite a bit by using the full convolutional kernel (which mixes feature maps across all channels).
>
> Regarding the ablation study, we think the idea of input injection is very interesting indeed, and can now see broader use in light of our results. However, concerning the use of dense weight matrices, we believe this is quite significant as well. Note that we controlled for model capacity in the ablative analysis: when we replace full weight matrices with sparse ones we are actually using LSTMs with the same number of parameters, which were the SOTA on PTB. A >2 perplexity improvement is a large improvement at this level of performance: e.g., the ICLR 2018 oral paper [1] improved upon [2] by 2.8 units of perplexity via MoS, and [3] improved upon [2] by 0.5 perplexity via extensive hyperparameter search. As another datapoint, on WikiText-103, generalizing from sparse weight (LSTM) to full weight (TrellisNet) leads to a significant improvement by about 6 units of perplexity (we use the best LSTM results reported, which is [4]); i.e. by 16%.
>
>
> [1] Yang, Zhilin, et al. "Breaking the softmax bottleneck: A high-rank RNN language model." arXiv preprint arXiv:1711.03953(2017).
> [2] Merity, Stephen, Nitish Shirish Keskar, and Richard Socher. "Regularizing and optimizing LSTM language models." arXiv preprint arXiv:1708.02182 (2017).
> [3] Melis, Gábor, Chris Dyer, and Phil Blunsom. "On the state of the art of evaluation in neural language models." arXiv preprint arXiv:1707.05589 (2017).
> [4] Rae, Jack W., et al. "Fast Parametric Learning with Activation Memorization." arXiv preprint arXiv:1803.10049 (2018).

---

### Official Review · AnonReviewer2 · 2018-11-06
**Review of "Trellis Networks for Sequence Modeling"**

**Rating:** 7
**Confidence:** 3

**Review:**

The authors propose a family of deep architecture (Trellis Networks ) for sequence modelling. Paper is well written and very well connected to existing literature. Furthermore, papers organization allows one to follow easily.  Trellis Networks bridge truncated RNN and temporal convolutional networks. Furthermore, proposed architecture is easy to extend and couple with existing RNN modules e.g. LSTM Trellis networks. Authors support their claims with an extensive empirical evidence. The proposed architecture is better than existing networks.
Although the proposed method has several advantages, I would like to see what makes proposed architecture better than existing methods.

---

> ### Author Response · Authors · 2018-11-16
> **Response to AnonReviewer2**
>
> Thank you for the positive feedback.
>
> We briefly recap the difference between TrellisNet and existing methods (specifically RNNs and TCNs). We showed that truncated RNNs are sparse TrellisNets with only weight parameters on the diagonal (see Eq. (5)). Essentially, once we replace the sparse weight with a dense weight matrix, there is no longer the interpretation of “RNN layers”. The off-diagonal parameters of the full, dense weight matrices mix hidden units at different history starting points, and there is no analog of this in recurrent networks. Similarly, for TCNs, the very idea of weight-tying and input-injection (inspired by RNNs in our work) is very unusual and has not (to the best of our knowledge) been used in existing temporal ConvNets on sequences. TrellisNet bridges both architectures, and is thus able to absorb architectural and regularization techniques from both sides.

---

### Author Response · Authors · 2018-11-16
**Revision to the paper**

We want to thank all reviewers for their feedback and suggestions. In order to address the comments in the reviews, we have updated our paper. The key changes in the revision are as follows:

1) We included a short introduction to temporal convolutional networks (TCN) at the beginning of Section 4.1.

2) We reorganized the experiment result tables in Section 5 for better clarity. Following the reviewers’ advice, we also include the performance of the generic TCN (from [1]) on the word-level PTB and char-level PTB tasks.

For the questions and the other interesting points that the reviewers brought up, such as the usage of full weights, we have included clarifications in our responses. We are happy to discuss further.

[1] Bai, Shaojie, J. Zico Kolter, and Vladlen Koltun. "An empirical evaluation of generic convolutional and recurrent networks for sequence modeling." arXiv preprint arXiv:1803.01271 (2018).

---

### Public Comment · (anonymous) · 2018-12-10
**Comparison to Yang et al., 2018**

In your paper, you report the result for Yang et al., 2018 without finetuning; with finetuning, they report a test perplexity of 54.44, which is slightly better than your reported result for the base TrellisNet with MoS of 54.67.  To make the comparison fair, could you report the number of epochs you train the TrellisNet for?  I believe Yang et al., 2018 train for 1000 epochs and then allow for another 1000 in the finetuning step as long as the loss keeps improving.  Thanks!

---

> ### Author Response · Authors · 2018-12-10
> **Epochs needed on PTB using MoS**
>
> Thank you for your interest in our paper!
>
> To obtain the 54.67 ppl on PTB using MoS, we trained for 400 epochs (similar for the 54.19 ppl result). We did not use finetuning step like Yang et al.
>
> In addition, the code will be made available so that you can run on your own as well :-)

---

### Meta-Review · Area_Chair1 · 2018-12-19
**An interesting novel approach combining advantages of truncated RNNs and temporal convnets**

**Confidence:** 4
**Recommendation:** Accept (Poster)

**Metareview:**

The paper proposes a novel network architecture for sequential learning, called trellis networks, which generalizes truncated RNNs and also links them to temporal convnets. The advantages of both types of nets are used to design trellis networks which appear to outperform state of art on several datasets.  The paper is well-written and the results are convincing.